# COVID-19 Risk Compensation? Examining Vaccination Uptake among Recovered and Classification of Breakthrough Cases

**DOI:** 10.3390/healthcare11010058

**Published:** 2022-12-26

**Authors:** Arielle Kaim, Gal Zeevy, Mor Saban

**Affiliations:** 1Department of Emergency & Disaster Management, School of Public Health, Sackler Faculty of Medicine, Tel-Aviv University, Tel-Aviv-Yafo 6139001, Israel; 2National Center for Trauma & Emergency Medicine Research, The Gertner Institute for Epidemiology & Health Policy Research, Sheba Medical Center, Tel-HaShomer, Ramat Gan 5266202, Israel; 3The Information & Computerization Unit, The Gertner Institute for Epidemiology & Health Policy Research, Sheba Medical Center, Tel-HaShomer, Ramat Gan 5266202, Israel; 4Health Technology Assessment and Policy Unit, The Gertner Institute for Epidemiology & Health Policy Research, Sheba Medical Center, Tel-HaShomer, Ramat Gan 5266202, Israel; 5Nursing Department, School of Health Professions, Sackler Faculty of Medicine, Tel-Aviv University, Tel-Aviv-Yafo 6139001, Israel

**Keywords:** COVID-19, immunity, reinfection, vaccination uptake, breakthrough cases

## Abstract

The study has two primary aims: the first is to examine the uptake of COVID-19 vaccination patterns among those previously infected, and the second is an evaluation of the period elapsed between the patient’s latest dose of the vaccine and the infection itself by demographic group. A retrospective study was conducted from 1 March 2020, to 31 May 2022, in Israel. The study found that among Israelis, vaccination uptake following infection is relatively low. When examining gender, one sees that the immunization rate among recovering females is higher than among men. Similarly, differences in uptake exist between age groups. When examining the interval between vaccine dose and infection according to age groups, the most significant breakthrough infection rate is among the ages of 20–59 (1–6 days—0.3%; 7–13 days—0.48%; two to three weeks—0.3%, *p* < 0.001). This study reveals potential reservoir groups of virus spread. Among previously infected, low vaccination uptake levels are observed (first dose—30–40%, second dose—16–27%, third dose—9% and fourth dose—2%, *p* < 0.001), despite findings that indicate surging reinfection rates. Among vaccinated, two critical groups (0–19; 20–59) exhibit highest levels of breakthrough cases varying per vaccine doses, with statistically significant findings (*p* < 0.001). These population groups may be subject to a false sense of security as a result of perceived acquired long-term immunity prompting low perceived risk of the virus and non-vigilance with protective behavior. The findings point to the possibility that individuals engage in more risky health behavior, per the Peltzman effect.

## 1. Introduction

Coronavirus disease 2019 (COVID-19) has exhibited the impact of a novel, infectious pathogen on all facets of life for the global community [1]. In December 2020, alongside the implementation of various countermeasures (e.g., social distancing, mask-wearing, mass testing, contact tracing, and mobility restrictions), the emergence of effective vaccines and the respective inoculation rollout campaigns were integrated in the fight against the pandemic, offering optimism for suppressing the virus transmission [2,3,4,5]. Findings have indicated that the combination of non-pharmaceutical interventions and vaccination would offer synergistic targeting of the virus [6,7]. The World Health Organization (WHO) has deemed vaccines as an essential step for management of the pandemic [3]. Acquired immunity on the individual level is established either through immunization with a vaccine or through natural infection with the pathogen [8]. While vaccines and non-pharmaceutical measures have shown effectivity in reducing morbidity and mortality rates from COVID-19, long-lasting flattening of the epidemic curve has not been reached to date. [9,10,11]. Immunity has also been challenged by virologically confirmed SARS-CoV-2 reinfection of previously infected individuals and vaccine breakthrough cases [10,11]. The challenge to the prospect of containment from fading immunity and reinfection has been an acknowledged concern since the beginning of the pandemic [12,13]. While immune responses are acknowledged to be heterogenous among individuals, findings from epidemiological analyses have reported natural immunity protection from reinfection for at least 6–12 months [14,15,16,17,18]. Reinfection cases are expected to occur when immunity wanes or the pathogen’s antigenicity evolves, resulting in immune evasion [11]. Uncertainty was initially expressed in messaging regarding whether those previously infected persons would benefit from vaccination [19]. Subsequent findings, for example, those from the UK, indicated that for a more efficient vaccine rollout, those previously infected subjects would need only one dose of a vaccine, where they saw that antibody titer rose in subjects by 140-fold [20]. The characterization of the effectiveness of acquired immunity from vaccination with the BNT162b2 vaccine in several studies has demonstrated modest rates of breakthrough infection and disease against the beta (B.1.351), delta (B.1.617.2) and omicron variants (B.1.1.529) variants, whereas other studies showed higher rates [21,22,23]. In addition, findings by Goldberg et al., indicated waning immunity a few months after receipt of a second inoculation dose [24].

Israel, a country of 9.3 million, navigated the pandemic by implementing three national lockdowns between March 2020 and January 2021. In addition, Israel rolled out a national vaccination campaign for two initial doses of the Pfizer BNT162b2 mRNA vaccine, as well as for a third “booster” dose after a fourth wave surge dominated by the delta (B.1.617.2) variant and a second “booster” dose during the Omicron variant surge. The success of a vaccination drive in containing the spread of the virus highly relies on the uptake levels of vaccination by the public [25,26]. Beginning on 20 December 2020, the first national vaccination drive began where in the first weeks, healthcare workers, individuals aged 60 years and above, and those considered to be at risk were invited to be vaccinated, where others were gradually added to the targeted populations [5]. Despite initial momentum in the Israeli vaccination drive, full public inoculation was hampered by vaccine skepticism [27,28]. Beginning on 13 July 2021, the first “booster” campaign was similarly offered to immunocompromised patients and gradually expanded to include the entire population over the age of 12 years old. Furthermore, vaccination of children of ages 5–11 began in November 2021. Similarly, in December 2021, the second “booster” began to be administered, but with even lower uptake rates among the public. Findings by Meng et al. regarding booster vaccination have indicated that the public does not need to uptake booster doses concurrently. Rather, the need for a booster may be delayed, with factors impacting antibody levels being age, gender, underlying diseases, and immunosuppressive treatments [7]. 

The challenges to vaccine uptake were addressed via a mix of incentives. Many of the measures addressed the general population, while others were more specifically targeted at the more challenging subgroups [29]. Most notoriously, a “green pass” certificate, which provides access to social, cultural, sports events, gyms, hotels, and restaurants as well as isolation exemptions (e.g., upon contact with a confirmed COVID-19 case or upon returning from international travel), was implemented with updated eligibility guidelines after the requirement for the “booster”, 6 months following the second inoculation dose [27]. According to Ministry of Health guidelines concerning those who were previously infected, initially COVID-19 recoverees were not eligible for vaccination during the first few months of the national campaign; however, this was later updated in October of 2021, where to be eligible for the “green pass” for recoverees, less than six months from infection must have passed or persons must have received one dose of the vaccine [29]. Vaccination in the second, third, and fourth doses for recovered patients is voluntary. 

In light of this information, in the current study we have two primary aims—The first is to describe the uptake of COVID-19 vaccination patterns among those who had previously been infected with the virus in the first year and a half of an available COVID-19 vaccine in Israel. The second is an evaluation of the period elapsed between the patient’s latest vaccine dose received prior to infection, to the infection itself by demographic group. Given that reinfection and the emergence of novel variants have challenged the management of the pandemic, it is essential to understand what sectors of the population are inadequately conferring immunity as case surges continue to be observed during the ongoing pandemic. 

## 2. Materials and Methods

A retrospective study was conducted from 1 March 2020, to 31 May 2022. The data file includes the entire population of Israel (*n* = 9,289,761 of which 4,613,239 are male and 4,676,522 are female). Data were obtained from the Israeli Ministry of Health’s (MOH open COVID-19 database [30], which includes information about recovered patients characteristics (age and gender) and their vaccination rates in the first, second and third vaccine doses. We also collected data regarding time elapsed from infection to each vaccine dose (adjusted to Israeli MOH guidelines). In total, 72.2% of the population received the first dose (*n* = 6,710,568); 66.1% received the second dose (*n* = 6,142,401); and 48.4% and 8.7% received the third dose (*n* = 4,240,819) and forth dose (*n* = 816,667), respectively. The MOH dashboard produced by the Ministry of Health, is a national and daily snapshot, which presents data about the spread of the COVID-19 virus in Israel and includes summary data—aggregated, not identified.

### Data Analysis 

Data about vaccination status of two subpopulations among recovered patients in Israel were collected. First, an examination of vaccination rates among recovered patients who did not receive any vaccine dose prior to infection was conducted. In each month from March 2020 to May 2022, we examined the vaccination rates in each vaccine dose (0, 1, 2, 3, and 4 doses) of all recovered patients from that month (dynamic cohort). Vaccination rates were compared by gender (54% female) and by the following age groups: children and young adults (0–19 years, 39.1%), adults (20–59 years, 50.2%), and the elderly (60 years and above, 10.7%). Non-vaccinated individuals were defined as those who had not received any vaccine (or were less than 1 week after their first dose). Second, for all recovered patients who were infected after vaccination, we measured the period elapsed between patient’s latest vaccine dose received prior to infection, to the infection itself. A few periods of time were defined for each dose. For the 1st dose, the periods examined were as follows: infection 1–6 days after vaccination, 7–13 days, 14–20 days, and 20 days or above. For the 2nd dose and the 3rd dose, the periods examined were infection 1–6 days after vaccination, 7–13 days, 14–30 days, 31–90 days, and 3 months or above. These data are unavailable for the fourth dose. We conducted a one-way ANOVA to assess for statistical differences between groups. Ultimately, we measured the rates of infected patients after receiving a specific dose from all patients who received it in the specific time periods. We also compared those rates among the age groups of children and young adults (0–19 years, 44%), adults (20–59, 45%), and elderly (60 years and above, 11%). 

Finally, we computed the re-infection percentage from the beginning of the pandemic until now among those vaccinated and unvaccinated. Data analysis was performed in SPSS software version 28.0.1.1 and python with Pandas, Numpy, and Matplotlib packages on PyCharm platform. *p*-values lower than 0.05 were considered statistically significant.

## 3. Results

Since the pandemic began at the end of December 2021, 4,202,599 (approximately 45.9%) Israeli people in Israel were detected to be positive for the COVID-19 virus. The rate of recovered patients from March who did not receive any vaccination until the study examination date stood at approximately 27%, and this percentage increased over the following months. The percentage receiving the first dose among those who contracted the virus during the first year of the pandemic was fairly stable and stood between 30–40%. Among those receiving the second dose, the percentage was relatively high among those infected in March 2020 (27%) but decreased as time went on, whereas, with those infected in October and November 2020, second dose inoculation stood at a rate of approximately 16%. Regarding the third dose of inoculation among this population, this rate stood at approximately 9% in March and decreased in the subsequent months. Regarding the fourth dose, approximately this rate stood at 2% in March of 2021 and significantly decreased in the following months. See Figure 1. In 2021, vaccination rates among all doses decreased drastically. Despite this trend, in the nearest months (March–May 2022), the uptake of the fourth dose among the most newly infected is between 40–55%. These findings are indicative of the notion that the population that already had received three doses of the vaccine were subsequently infected in the nearest months and have decided to receive the fourth dose.

When examining gender, one sees that the rate of immunization among recovering females is higher than among men (See Figure 2). In most months in 2020, the percentage of men with 0 doses is higher than woman, but not by much, while immunization rate of dose 1 was higher among women by 5% more in most months. There are no substantial differences in vaccination rates in the second, and third doses, despite women receiving both doses at higher rates. However, among recoveries in recent months, much larger differences in favor of uptake of vaccination among women are observed regarding the fourth dose.

Dose immunization rates are relatively low in young people 0–19 years old compared to older ages, 20–59 years and above the age of 60 (See Figure 3). A similar trend was observed in the second and booster doses (third and fourth) of vaccines, where response rates to immunization were significantly higher among the older population. 

When examining the interval between vaccination and infection according to age groups, it can be seen that relative to the population, the rate of infection in the near period (1–13 days) after first dose inoculation is higher among the young population. A higher percentage of infections occurred between 1–6 days among young age groups, 0.6% in ages 0–19 years old and 0.3% in ages 20–59 years old as compared to 0.17% in ages 60 years and above (*p* < 0.001). Similarly, a higher percentage of infections between 7–13 days among young age groups, 1.3% in ages 0–19 years old, and 0.48% in ages 20–59 years old compared with 0.4% can be observed in ages 60 years and above (*p* < 0.001). Within the infection range of two to three weeks, the prevalence of infection in the oldest population group stands at 0.32% compared with the age groups 20–59 years (0.3%) and 0–19 years (1.3%) (*p* < 0.001).

When examining dose 2, one sees that in the long run, 3 months after vaccination, the rate of infection is highest among the population group aged 0–19 years, where approximately 12% of all those vaccinated in the second dose are infected 3 months or more after receival (*p* < 0.001). Similarly, the rate for the age group of 20–59 years stands at 8.5% among those vaccinated in the second dose who are infected 3 months or more after receival. The infection rate in the window of time between two weeks to three months is also higher among the youngest age group (0–19 years) (*p* < 0.001). 

When investigating the infection after the third booster dose, the rate of infection is higher among the age group of 20–59 years after three months (24.5%) as compared to 60 years and above (18.5%) and those 0–19 years (8.5%) (*p* < 0.001). The rate of infection after 3 months of receiving the vaccine increases significantly among all age groups compared to the previous time period, and in the age group of 20–59 years in particular (See Figure 4).

Figure 5, which indicates re-infection, shows that from the beginning of the epidemic until now, the re-infection rate is about 6.22% of all recoverees. This rate is significantly higher among those who are not vaccinated compared to those who have been vaccinated, at least in one dose.

## 4. Discussion

The results of this study offer several interesting findings regarding vaccine uptake among those recovered from COVID-19 throughout the first 18 months since the beginning of a national vaccination drive in Israel, alongside breakthrough infection rate comparisons by age group and time since inoculation. The study found that among Israeli residents, uptake of vaccination following infection is relatively low. While scientific understanding concerning natural-infection-derived immunity is continuously emerging regarding duration of protection and response to novel variants of concern, findings and guidelines have corroborated that vaccination can provide improved protection for previously infected persons [31]. Prior studies have identified previous infection as being associated with lower intention for vaccination and vaccine hesitancy; however, previous epidemiological evidence has not been documented on the subject per the authors knowledge [32,33,34]. Findings of Kaim et al. indicated that those who were previously infected were also less careful about additional health protective behavior, such as mask-wearing and social distancing, rendering them a significant potential reservoir of spread [26]. Individuals who have previously been infected and successfully recovered from the virus may have a diminished sense of perceived risk and perceived severity of the virus as exposure has already been confronted. It is discussed in the literature that induction of fear is usually the case in the context of novel risks; however, repeated exposure may result in a diminished arousal of fear, thus resulting in risk underestimation and laxer risk behavior [35]. Furthermore, given that throughout the early stages of the pandemic, critical answers concerning the course of immune response for protecting the individual from reinfection were delayed, and recognition of potential reinfection was only later established as a phenomenon that may have congruently contributed to the lower levels of perceived risk among this population and vaccination uptake. Consistent with additional findings on vaccine hesitancy, younger age was observed as a predictor for lower vaccine uptake among those previously infected [36,37,38]; however, regarding our findings on vaccine uptake following previous infection and gender, they were inconsistent with the literature, which documented lower uptake observed among women [36,39]. As mentioned in Robertson et al., women have previously been documented to be more vaccine-hesitant, with women more likely stating that their main reason for hesitancy is concern of side effects and lack of trust in vaccine [37]. However, in support of the current results in Israel, Lazarus et al. indicated in a global survey that men were slightly less likely to respond positively than women to potentially accepting the COVID-19 vaccine if offered [40].

Side effects of the COVID-19 vaccine have been widely examined. The concern with side effects of the vaccine has been widely documented throughout the pandemic, where most common side effects have included pain, redness or swelling at the site of vaccine shot, fever, fatigue, headache, muscle pain, nausea, vomiting, itching, chills, and joint pain, as well as rare cases of anaphylactic shock [41]. The findings of Lounis et al. have indicated that side effects to booster vaccination were more severe as compared to the primer doses [42].

The trends in the rates of breakthrough infection data similarly point to the younger populations of (0–19 years) and (20–59 years) as serving as a key source of infection spread, despite the existence of data concentrating on those who have decided to get vaccinated. The findings point to the possibility that individuals in this group similarly become less vigilant about protective behavior and engage in more risky behavior, as a false sense of security from vaccination may arise, per the Peltzman effect [26]. Previous findings have documented that older populations were previously found to engage in more self-preservatory behavior as they were more likely perceive COVID-19 as a significant crisis [43]. 

The findings further elucidate “vaccine fatigue” among the Israeli general public, where despite new variant scares and an ongoing sixth wave, people’s vaccination intention is compromised largely due to the needs to balance the burden and burnout associated with vaccination for COVID-19 alongside the feelings of duty, solidarity and social conscience [44]. As talks in Israel are underway regarding a fifth dose to tackle the sixth wave spike, an important question to consider is whether if rolled out and even incentives will be implemented, will further vaccination be relevant given the observed current uptake of previous booster doses and the breakthrough infection rates observed. Outside of COVID-19 vaccination, it is further observed that routine vaccinations are similarly being impacted by the vaccine fatigue as the intense focus over the past two years on COVID-19 and the development of COVID-19 vaccines has been to the almost exclusion of other infections and vaccines [45]. In order to avoid long-term detriment to the relationship of the public with public health measures, this is a cycle that needs a novel solution and perspective.

The limitations of this study include the fact that the sixth pandemic wave in Israel is still ongoing, and it reduces the ability to forecast the attributes of the “remaining” period of this wave. Furthermore, because the cohort is dynamic with a very large and heterogenous population, it is difficult to define the precise reason for non-immunization. In addition, the severity of the disease in each individual is unknown, which may be an influencing element on risk perception. We did not examine the attitudes and perceptions of recovering patients toward vaccination. Despite these limitations, this study, through a longitudinal, epidemiological assessment of vaccination uptake among recovered and a breakthrough case evaluation, reveals potential reservoir groups of virus spread. In addition, the study makes use of a national dataset since the beginning of the pandemic. Future research should aim to assess factors and strategies that could influence vaccination uptake among these population groups.

## 5. Conclusions

Considering the findings of this study and indications of reinfection rates in Israel are on the rise with new variants arising, targeted messaging should be directed towards the relevant population groups for emphasizing the necessity of continued vigilance in behavior, including the uptake of vaccination for those previously infected and continued mask-wearing and physical distancing for those fully vaccinated and of younger age. These findings have generalizable and valuable implications for authorities and health care providers in identifying groups with lower vaccination compliance and lower levels of vigilance in protective health behavior in the context of the current pandemic. To ensure the public’s compliance with health directives and continued curtailing of the pandemic, guidelines and information campaigns must reemphasize that the efforts against the COVID-19 war are not yet over.

## Figures and Tables

**Figure 1 healthcare-11-00058-f001:**
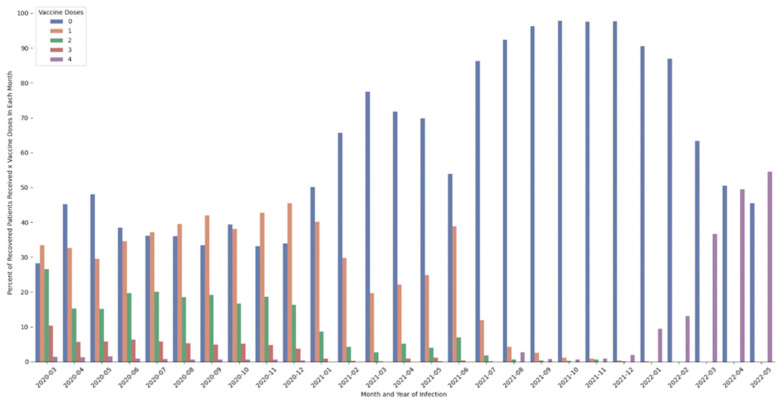
Per-month infection (year-month), percentage of recovered patients according to number of vaccine doses received. Note: data only include recovered patients who received no vaccine doses prior to infection.

**Figure 2 healthcare-11-00058-f002:**
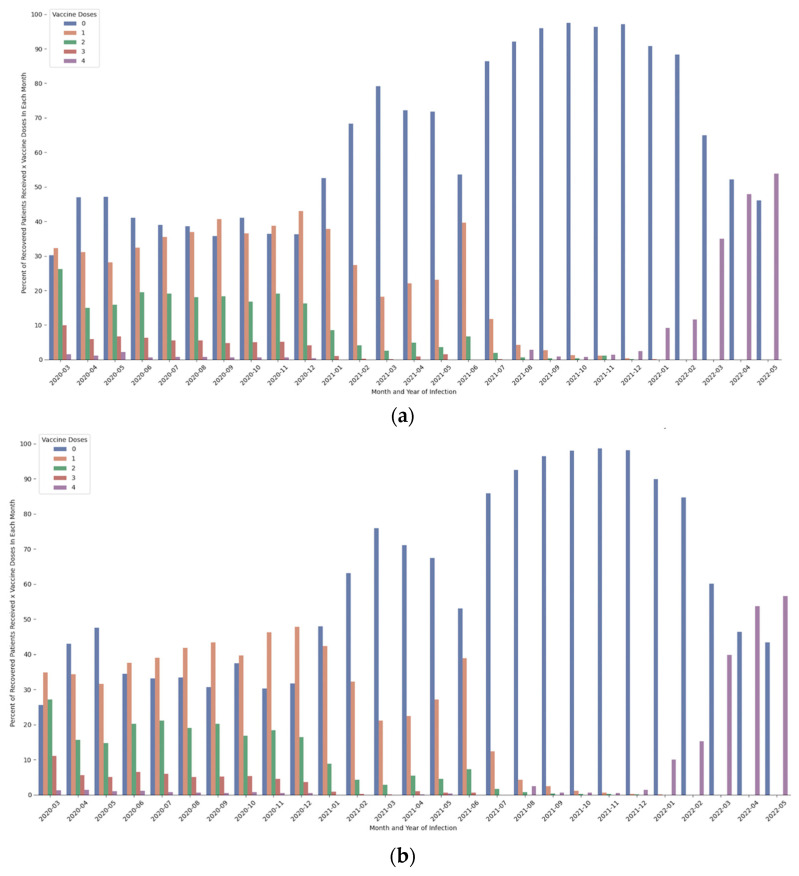
Per-month infection (year-month), percentage of recovered patients by gender ((**a**)—male individuals; (**b**)—female individuals) according to number of vaccine doses received. Note: data only include recovered patients who received no vaccine doses prior to vaccination.

**Figure 3 healthcare-11-00058-f003:**
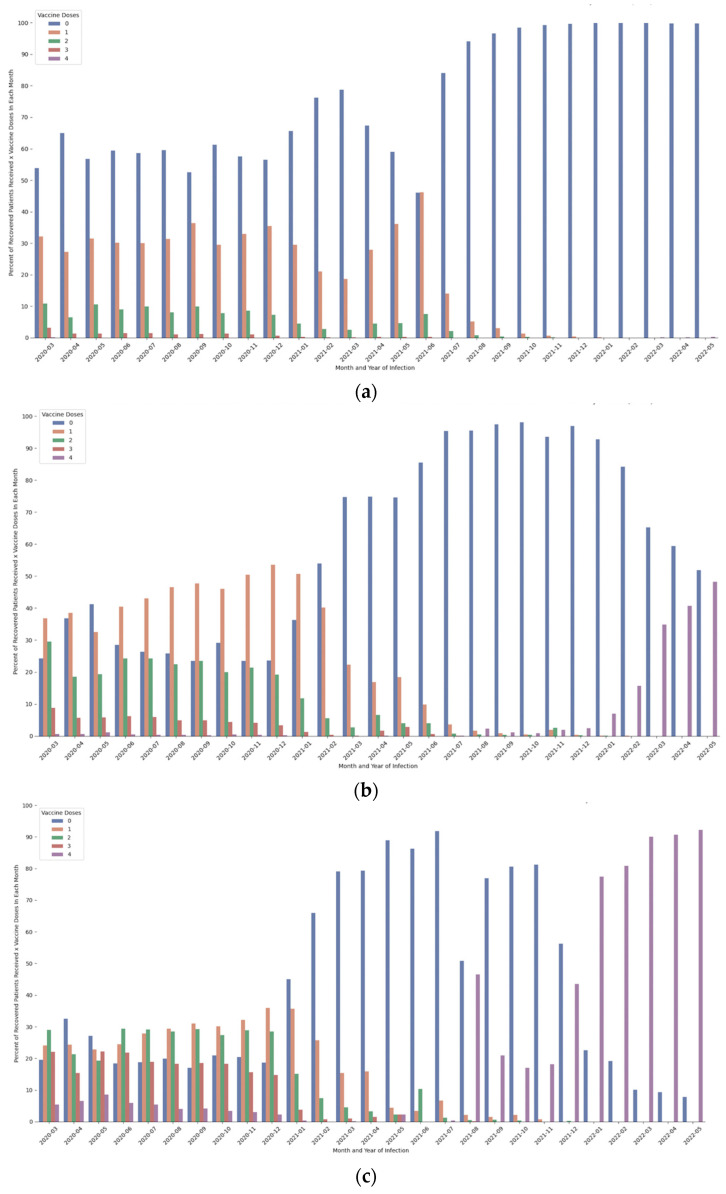
Per-month infection (year-month), percentage of recovered patients by age ((**a**): 0–19 years; (**b**): 20–59 years; (**c**): 60 years and above) according to number of vaccine doses received. Note: data only include recovered patients who received no vaccine doses prior to infection.

**Figure 4 healthcare-11-00058-f004:**
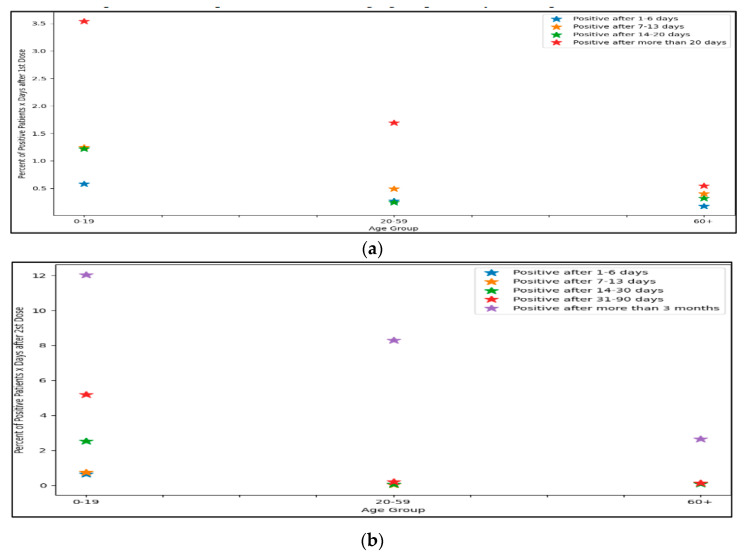
Percentage of positive patients per age group after X (denoted in legend) days after vaccination dose ((**a**): first dose; (**b**): second dose; (**c**): third dose).

**Figure 5 healthcare-11-00058-f005:**
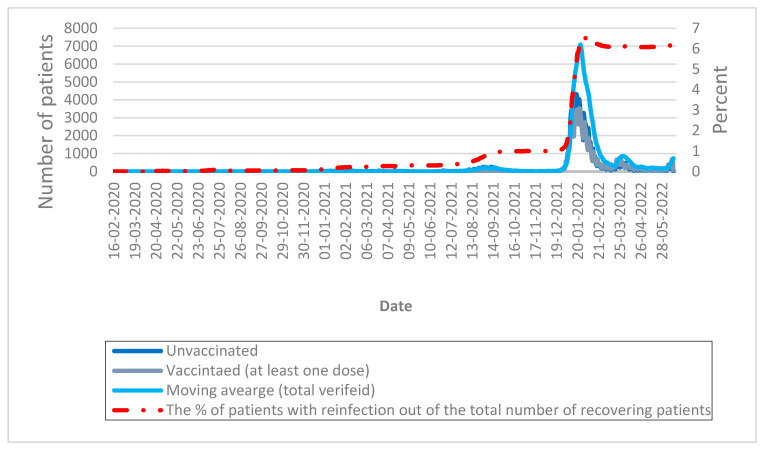
Percentage of patients with reinfection of the total number of recovering patients by date (with vaccination status denoted).

## Data Availability

Links to publicly archived dataset: https://datadashboard.health.gov.il/COVID-19/general (accessed on 1 June 2022).

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
