# Peer review of "COVID-19 Risk Compensation? Examining Vaccination Uptake among Recovered and Classification of Breakthrough Cases"

_healthcare, 2022, doi:10.3390/healthcare11010058_

Round 1
Reviewer 1 Report
Congratulation to all authors for such an interesting and important retrospective research paper in Israel which is crucial for the stakeholders i.e. policymakers and healthcare providers. I would suggest the following to improve the manuscript:
· The findings section gender-wise analysis shows that females have higher updates of both doses but this has not been discussed in the discussion section. Why do females have the higher update in Israel.
· Figure 1 – 3C should be the reader-friendly which are not very clear in the current formats,
· Use 60 years and above rather than 60+,
· Check line 287 – persons28 which may need to fix the citation. Correct this.
· The authors have talked about the limitations of the study but it would also be great to talk about the strengths of this study.
· Authors should recommend based on this study's findings to different level stakeholders i.e. Israel policy level/health care providers/researchers under the conclusion section where authors concluded the study within a single sentence.
· The reference section needs to be a thorough review (i.e inconsistent references – 1-3 and others below) and needs to adhere to the journal format.
Reviewer 2 Report
Dear authors:
The research point is very good. However, there are a few comments to the authors:
1. In the abstract: the percent and the p-value should be mentioned.
2. Data analysis: why you did not use any statistical test to compare groups
3. All over the manuscript the word (years) should be mentioned with the age groups.
4. Did infants (0-6 years) received the vaccine??
5. Children are not from 0 to 19 years old.
6. Figures 1, 2, and 3 are not clear
7. Better to add a table comparing all groups at the different periods
8. Other comments are within the manuscript file

Reviewer 3 Report
I reviewed the paper "COVID-19 Risk Compensation? Examining vaccination uptake 2 among recovered and classification of breakthrough cases" submitted for publication in "Healthcare" journal.
The paper is interesting the study was well conducted. However, there are some improvements that must be done to merit publication in this journal:
1. line 16-17 please rephrase, delate "those" and write among "patients" or "individuals" previously infected
2. Please include more results (numbers) in your abstract0
3. Line 31, it's COVID-19
4. Line 34, please write the full name: Coronavirus disease 2019 (COVID-19) as it is the first time it appears.
5. Line 35-39. There is a lack of references in the first part of your introduction, your background should be more referenced, here are some references to add:
* https://doi.org/10.1016/j.biopha.2021.112015
* https://doi.org/10.3390/healthcare10071341
* https://doi.org/10.1080/17512433.2021.1902303
6. Line 61, please delate 2021.
7. It's better if you write at least 1-2 sentence about the side effects of vaccination, specifically booster doses, here are some papers to add:
*https://doi.org/10.1080/20009666.2021.1974665
*https://doi.org/10.3390/vaccines10111781
*https://doi.org/10.26355/eurrev_202102_24877
8. Line 135-137, please rephrase to avoid redundancy.
9. Please consider improving the overall quality of your figures, some of them are very difficult to read or even unreadable.
10. Figure 5, it should contain X-axis and Y-axis
11.Line 287, 28 ???
12.Line 289, please remove 2021
13. In your discussion section, please include more results and compare them with other studies, it's hard to detect what your study adds to current knowledge.
14. Please rephrase the first part of your conclusion as it is not representative of your study.
Round 2
Reviewer 3 Report
I would like to thank the authors for the efforts they put in this revision, the overall level of the manuscript was highly improved.
I would also like to thank them for providing detailed responses to reviewers and for highlighting the improvements using track changes. That was very clear.
I just have a very minor comment, the references list is still a little bit lacking, I suggest to add these references concerning preventive measures and vaccination:
*https://doi.org/10.1111/1751-7915.13997
*https://doi.org/10.3390/v14122771
*https://doi.org/10.1002/jmv.27590
Apart from that, I think the manuscript is now suitable for publication.
Best wishes!
